# Finger Vein Recognition Based on Unsupervised Spiking Convolutional Neural Network with Adaptive Firing Threshold

**DOI:** 10.3390/s25072279

**Published:** 2025-04-03

**Authors:** Li Yang, Qiong Yao, Xiang Xu

**Affiliations:** Artificial Intelligence and Computer Vision Laboratory, Zhongshan Institute, University of Electronic Science and Technology of China, Zhongshan 528402, China; 202221080821@std.uestc.edu.cn (L.Y.); xuxiang@zsc.edu.cn (X.X.)

**Keywords:** finger vein recognition (FVR), adaptive firing threshold, spiking convolutional neural network

## Abstract

Currently, finger vein recognition (FVR) stands as a pioneering biometric technology, with convolutional neural networks (CNNs) and Transformers, among other advanced deep neural networks (DNNs), consistently pushing the boundaries of recognition accuracy. Nevertheless, these DNNs are inherently characterized by static, continuous-valued neuron activations, necessitating intricate network architectures and extensive parameter training to enhance performance. To address these challenges, we introduce an adaptive firing threshold-based spiking neural network (ATSNN) for FVR. ATSNN leverages discrete spike encodings to transforms static finger vein images into spike trains with spatio-temporal dynamic features. Initially, Gabor and difference of Gaussian (DoG) filters are employed to convert image pixel intensities into spike latency encodings. Subsequently, these spike encodings are fed into the ATSNN, where spiking features are extracted using biologically plausible local learning rules. Our proposed ATSNN dynamically adjusts the firing thresholds of neurons based on average potential tensors, thereby enabling adaptive modulation of the neuronal input-output response and enhancing network robustness. Ultimately, the spiking features with the earliest emission times are retained and utilized for classifier training via a support vector machine (SVM). Extensive experiments conducted across three benchmark finger vein datasets reveal that our ATSNN model not only achieves remarkable recognition accuracy but also excels in terms of reduced parameter count and model complexity, surpassing several existing FVR methods. Furthermore, the sparse and event-driven nature of our ATSNN renders it more biologically plausible compared to traditional DNNs.

## 1. Introduction

With the escalating demand for enhanced security, traditional identification technologies, such as cards and passwords, are progressively being supplanted by biometric traits. Biometrics, encompassing face [1], voice [2], gait [3], fingerprint [4], finger vein [5], and palm vein [6], among others, have garnered considerable attention due to their inherent security, reliability, and myriad advantages. Broadly speaking, these biometrics can be categorized into two folds: extrinsic and intrinsic. Extrinsic biometrics, including face, voice, gait, and fingerprint, are susceptible to vandalism and forgery due to their exposed nature. Additionally, they are vulnerable to environmental factors; for instance, poor lighting or background noise can significantly impair identification accuracy. Given these constraints, intrinsic biometrics offer greater convenience and reliability in practical applications. Among them, finger vein (FV) traits emerge as the most promising intrinsic biometric modality, as they are uniquely concealed beneath the skin surface. FV patterns are imaged through the absorption of near-infrared light (700–900 nm) by hemoglobin in the blood vessels, creating dark shadows that reveal the vein morphology [7]. Typically, FV acquisition devices are designed to be compact and non-contact, making them more convenient and hygienic to use. Beyond that, FV images can only be captured from a living individual, rendering them highly resistant to forgery or theft.

Currently, researchers have achieved noteworthy advancements in finger vein recognition (FVR) methodologies, which can be broadly classified into two categories: hand-crafted feature extraction and deep feature learning. Within the realm of hand-crafted feature extraction, various paradigms of feature representation have been employed: (1) Vein Features: In this approach, finger veins are initially segmented from the background, and specific vein patterns are utilized for recognition [8,9,10]. (2) Point Features: These methods involve extracting distinctive and representative points, such as minutiae points [11] and corner points [12,13]. Instead of measuring the similarity between entire images, these techniques measure the similarity between these extracted points, thereby simplifying the recognition process. (3) Statistical Features: These features are derived automatically through statistical analysis of large datasets. Examples include principal component analysis (PCA) [14], histograms [15], and sparse representation [16]. (4) Texture Features: This category encompasses techniques that capture the texture characteristics of finger veins. Techniques such as local line binary pattern (LLBP) [17], weighted local binary pattern (WLBP) [18], and even-symmetric Gabor filter [19] have been employed to extract texture-based features. In general, while hand-crafted feature extraction methods have demonstrated certain efficacy in FVR, they tend to have lower generalization capabilities and are more sensitive to variations in image quality, ambient lighting conditions, finger rotation, and shift. These limitations highlight the need for further research and development in this area, particularly in the context of deep feature learning methodologies.

Compared to hand-crafted methods for FVR, deep neural networks (DNNs), particularly convolutional neural networks (CNNs) and Transformers, have exhibited superior generalization capabilities and high-level semantic representation [20,21]. DNNs typically offer an end-to-end recognition process, facilitating continuous learning and parameter optimization based on the provided samples. In [22], a pre-trained VGGNet-16 was utilized for FVR, yielding impressive performance. Similarly, AlexNet was introduced in [23] to extract features from FV images. Furthermore, in [24], a three-channel composite image, as opposed to the conventional differential image, was input into DenseNet to enhance FV identification.

The large-scale DNNs mentioned earlier rely heavily on a sufficient quantity of training samples, which poses a challenge in practical FVR scenarios. To mitigate this issue, several lightweight models [25,26,27] specifically designed for FVR have been developed [28]. In [29], a lightweight dual-channel Siamese network was explored for FVR, aiming to reduce complexity while maintaining good performance. In [30], Gabor convolutional layers were introduced to decrease parameter complexity. Furthermore, in [31], a lightweight CNN incorporating a convolutional block attention module (CBAM) was proposed for FVR, thereby enhancing the network’s representation power by focusing on important features. Additional, in [32], skip connections in DenseNet were further refined to reduce memory overhead and create a more streamlined structure tailored for FVR.

Although the aforementioned lightweight models alleviate the demand for extensive training samples in FVR, it is crucial to acknowledge that these DNNs lack biological plausibility. Specifically, DNN neurons employ static and continuous-valued activation functions operating on a common clock cycle, whereas biological neurons utilize discrete spikes to propagate information asynchronously. This disparity leads to the development of spiking neural networks (SNNs), a third-generation neural network paradigm inspired by biological information processing. SNNs leverage sparse and asynchronous binary signals processed in a massively parallel manner, with individual spike signals temporally sparse-coded to represent the occurrence of specific events via precise action potentials. This makes SNNs more biologically realistic than DNNs. Furthermore, SNNs exhibit low power consumption, rapid inference, and event-driven information processing due to their biologically plausible local learning rules and efficient neuromorphic hardware platforms. These virtues position SNNs as promising candidates for efficient DNN implementations, with various SNN architectures proposed for pattern recognition tasks. One approach involves converting pre-trained DNNs into deep SNNs to circumvent the limitations of gradient descent in SNNs [33,34,35,36]. In this scenario, activations of DNN analog neurons are translated into firing rates of spiking neurons. However, it should be noted that not all DNNs can be easily converted into SNNs, and firing rate-based conversion is suboptimal for SNNs. Given the significance of precise timing in biology, another approach focuses on temporal coding in conjunction with biologically inspired learning rules, such as spike timing-dependent plasticity (STDP) [37] or Hebbian learning [38,39,40]. Although these local learning rules allow for the detection of spatio-temporal patterns as features for data-efficient training, supervised error signals backpropagation remains challenging, resulting in accuracies inferior to those achieved by traditional DNNs trained with backpropagation. In addition, a growing body of research is exploring hybrid paradigms that integrate spiking neural networks (SNNs) with various advanced architectures, including machine learning models [41], generative adversarial networks [42], recurrent neural networks [43], convolutional neural networks [44], and self-supervised learning frameworks [45]. This cross-architectural fusion strategy significantly enhances the practical performance of SNNs in complex tasks by combining the advantageous features of different models. However, this hybrid approach also faces two critical limitations: first, the incorporation of non-spiking components compromises the biological plausibility of the model; second, the coupling of multiple architectures leads to nonlinear growth in computational complexity. Consequently, systematic trade-off optimization between model performance and implementation costs must be carefully considered during practical engineering deployment.

Drawing inspiration from the distinctive properties of SNNs, we propose a novel FVR approach based on an unsupervised spiking convolutional neural network (SCNN). Our proposed SCNN architecture comprises multiple cascading spiking convolutional layers and pooling layers, where information flows in the form of temporally coded spike signals. To preprocess the FV image samples, we employ Gabor and difference of Gaussian (DoG) filters, which emphasize orientation and contrast features, respectively. Subsequently, we utilize intensity latency encoding to transform these filtered images into spiking-encoded inputs, thereby enriching them with temporal information.

Given the non-differentiable nature of temporally coded spiking signals and the limitations of standard STDP rules in determining an optimal firing threshold, we have specially designed an adaptive firing threshold-driven STDP learning rule tailored for weight updating in our model. Ultimately, we retain the output spiking features with the earliest emission times and feed them into a support vector machine (SVM) for classification. Our model, termed the Multi-layer Unsupervised Spiking Convolutional Neural Network with Adaptive Firing Threshold (dubbed ’ATSNN’), leverages the discrete spike encoding of FV images to discern spatio-temporal dynamic features. In a nutshell, The main innovative contributions of our work are summarized as follows:First, we pioneer the application of SNNs in FVR, and introduce an unsupervised spiking convolutional neural network. This network not only achieves commendable performance on par with state-of-the-art FVR methods but also boasts a streamlined parameter set and reduced model complexity.Second, we incorporate an adaptive firing threshold mechanism that mimics the dynamic threshold behavior of biological neurons. This eliminates the need for manual threshold adjustments and enhances compatibility across diverse FV datasets. Additionally, the inclusion of batch normalization layers (BNs) after each spiking convolutional layer strengthens the training stability of the model.Third, by leveraging Gabor and DoG filters, we enrich the multi-directional edge features and multi-scale texture features of the FV images. This multi-filter strategy diversifies spike encoding, ensuring that both contrast and orientation information are captured effectively.

It is noteworthy that the proposed algorithm in this study primarily focuses on finger vein classification. Compared to traditional convolutional neural network models, this architecture demonstrates significant advantages in parameter efficiency and computational efficiency. The lightweight characteristics of the model make it particularly suitable for deployment on resource-constrained embedded recognition devices. In practical application scenarios, this model can serve as the foundation for building high-performance finger vein authentication systems.

The remainder of this paper is organized as follows. Section 2 provides a detailed elaboration on the entire identification process of our proposed ATSNN. Section 3 presents a comprehensive discussion of the experimental results, assessing the performance of our ATSNN model using three widely recognized finger vein databases: MMCBNU_6000 database [17], FV-USM database [46], and SDUMLA-HMT database [47]. Finally, Section 4 concludes the paper with some remarks and proposes potential avenues for future research.

## 2. Proposed ATSNN Model

In this section, we delve into the newly conceived ATSNN model within the context of finger vein recognition. Firstly, we present a broad overview of the fundamental workflow of the ATSNN, outlining its operational sequence in a clear and concise manner. Subsequently, we elaborate on the architecture design of the ATSNN model, focusing on the intricate details of its pivotal component blocks. We then provide detailed explanations of the local training mechanism and the adaptive threshold strategy employed in each spiking convolutional layer. Finally, we illustrate the classification and recognition process, specifically detailing how features are selected from the spike-coded output of the ATSNN for classifier learning.

### 2.1. FVR Workflow of the ATSNN

Figure 1 illustrates the fundamental workflow of finger vein recognition utilizing the ATSNN. In the preprocessing stage, the original ROI image is resized to an appropriate size (specifically, a unified size of 48×48 pixels in our experiments) and converted to grayscale, with intensity values ranging from 0 to 255. Subsequently, the converted gray-level image undergoes filtering using predefined DoG and Gabor filters to comprehensively extract image contrast and edge information. The specific filter settings are depicted in Section 2.1.1. To retain the most salient edges and contrast information, the filtered image maps are further processed by applying predefined thresholds for each type of filter. This thresholding step alleviates the negative effects of low-intensity signals on subsequent spike coding. Specifically, pixels with output intensity signals below a certain threshold (e.g., 50) are set to zero intensity value.

In the conversion stage of spike coding, two crucial steps are performed prior to encoding conversion: local normalization and pointwise inhibition. Local normalization serves the purpose of enhancing the local contrast within each feature map of the filtered image, thereby improving the discriminability of features. Specifically, the intensity values of each local spatial region are divided by the regional mean value. This process ensures that significant local features are accentuated while suppressing global variations in brightness, making the features more distinguishable for subsequent processing in the network. On the other hand, pointwise inhibition is applied across all image channels to ensure that, at most, one neuron (specifically, the neuron with the maximum pixel intensity value) in the same spatial position is allowed to fire. Pointwise inhibition enforces sparsity in cross-channel responses at the same spatial coordinates within feature maps by suppressing redundant responses from non-dominant channels, thereby compelling the network to establish orthogonalized feature representations across different channels. This hard selection mechanism biologically mimics the lateral inhibition effect observed in retinal ganglion cells, where inhibitory interactions among spatially neighboring neurons enhance contrast and sharpen edges in visual inputs. This process aligns with the energy optimization principles of biological visual systems, ensuring efficient and effective feature extraction.

Subsequently, intensity latency encoding is employed to convert the intensity values of pixels into spike sequences, represented in the form of spike bins. The specific details of the spike coding process are outlined in Section 2.1.2. Once the spike coding expression of the input image is obtained, these spike signals are sequentially fed into the ATSNN for model training and feature extraction. The specific architectural design of ATSNN is presented in Section 2.2. Given that ATSNN adopts a local unsupervised training mode, it only requires the spike signal of each image sample, without the corresponding class label information. After the network is thoroughly trained, it can be utilized to extract spike features. In summary, the input and output of ATSNN consist of spike signals with temporal coding.

Finally, in the classification stage, the output spike features of ATSNN are accumulated along the temporal dimension. The temporal slice features with the maximum cumulative value are selected for classifier learning. To train an SVM classifier in this context, the class label corresponding to each feature signal is required.

#### 2.1.1. Filter Settings

To extract salient features from an image for spike coding, the selection of appropriate filter types is crucial. Furthermore, the integration of diverse filters serves as an effective strategy to comprehensively capture various image features.

In the visual system, receptive fields delineate neurons’ capacity to perceive specific regions of the visual field. Typically, these receptive fields are organized in a center-surrounding configuration, where the center corresponds to the neuron’s most sensitive area for stimuli, while the surrounding regions exhibit weaker responses. To closely emulate the center-surrounding characteristics of retinal ganglion cells in the visual cortex and detect positive-negative or negative-positive differences in the image, the DoG filter is employed. This filter computes the difference between two Gaussian kernel functions with distinct variances, thereby accentuating edge and detail information in the image [48]. Neurobiological studies have revealed the presence of two types of neurons in the mammalian visual cortex: simple cells and complex cells. Simple cells are responsible for sensing edge and contrast information, while complex cells are responsible for sensing the orientation of image contours. The functionality of complex cells can be modeled using a pair of simple cells. To mimic the sensing capabilities of simple cells in terms of orientation and scale characteristics, a 2D Gabor filter is constructed by modulating a Gaussian function with a complex waveform, achieving locally optimal characteristics in both the spatial and frequency domains [49,50,51].

However, each filter type has inherent limitations. While the DoG filter boasts high computational efficiency and can swiftly locate edges and feature points in the image, it lacks frequency domain direction selectivity and is highly susceptible to noise. Conversely, the 2D Gabor filter excels at capturing directional changes in the frequency domain and is insensitive to illumination variations, but its computational complexity is relatively high. Therefore, by combining DoG and Gabor filters, we can more effectively extract image features. Specifically, DoG filters can be utilized to emphasize the high-frequency components (edges) of the image, followed by Gabor filters to further refine edge information and provide additional edge direction details.

Based on the analysis presented above, we utilize Gabor filters with varying scales and orientations, as well as DoG filters with different contrasts, as illustrated in Figure 2. Specifically, the four Gabor filters are configured with window sizes of (15×15) pixels and orientations of 3π/8, 5π/8, 7π/8, and 9π/8 radians. The Gabor filters are mathematically defined by Equations (Equation 1)–(Equation 3).(1)$$G\left(x,y;\lambda ,\theta ,\varphi ,\sigma ,\gamma \right)=\exp\left(-\frac{X^{2}+\gamma^{2}Y^{2}}{2\sigma^{2}}\right)\cdot \cos\left(2\pi \frac{X}{\lambda }+\varphi \right).$$(2)$$\begin{array}{cc} X & =x\cos\theta +y\sin\theta \\ Y & =-x\sin\theta +y\cos\theta . \end{array}$$(3)$$\hat{G}\left(x,y;\lambda ,\theta ,\varphi ,\sigma ,\gamma \right)=\frac{G\left(\cdot \right)-G{\left(\cdot \right)}_{\mathrm{mean}}}{G{\left(\cdot \right)}_{\max}}.$$

For practical application in finger vein recognition, only the real part of the Gabor function is employed. Within the Gabor function, (x,y) represents the spatial position coordinates of the pixels, σ denotes the standard deviation of the Gaussian envelope, λ is the wavelength of the cosine factor, θ specifies the orientation of the normal to the parallel stripes, φ signifies the phase offset of the cosine factor, and γ is the spatial aspect ratio. The output of the Gabor filtering, denoted as G(x,y;λ,θ,φ,σ,γ), captures the orientation information of the image. The normalized output of the Gabor filter is represented as $\hat{G}$.

Furthermore, the two DoG filters have a kernel size of 7×7 pixels. Filters with standard deviations of $\delta_{1}=1$ and $\delta_{2}=2$ mimic excitatory inputs for on-center ganglion cells, whereas filters with standard deviations of $\delta_{1}=2$ and $\delta_{2}=1$ emulate inhibitory inputs for off-center ganglion cells. The corresponding Gaussian difference algorithm is formulated in Equations (Equation 4) and (Equation 5).(4)$$D\left(x,y,\delta_{1},\delta_{2}\right)=\frac{1}{2\pi {\delta_{1}}^{2}}\exp\left(-\frac{x^{2}+y^{2}}{2{\delta_{1}}^{2}}\right)-\frac{1}{2\pi {\delta_{2}}^{2}}\exp\left(-\frac{x^{2}+y^{2}}{2{\delta_{2}}^{2}}\right),$$(5)$$\hat{D}\left(x,y,\delta_{1},\delta_{2}\right)=\frac{D\left(\cdot \right)-{D(\cdot )}_{\mathrm{mean}}}{{D(\cdot )}_{\max}}.$$

Here, $\delta_{1}$ and $\delta_{2}$ represent two distinct standard deviations of the Gaussian distribution, (x,y) are the spatial position coordinates of the pixels, and $D(x,y,\delta_{1},\delta_{2})$ represents the output of DoG filtering, which encapsulates the contrast information of the image. The normalized version of *D* is denoted as $\hat{D}$.

#### 2.1.2. Spike Coding

Once the filter conversion of the original ROI image is completed, the spike coding phase commences. Given that the form of neuronal firing is constant, with variations only in the number and timing, we adopt an intensity latency encoding strategy that incorporates both the actual pixel intensity and the temporal of spike firing, which meaning, the stronger the external visual stimulus is received (corresponding to the larger the pixel intensity), the earlier the spike emission time. Specifically, we segment all image pixels into a series of predefined time-steps. For an input image of dimensions F×H×W, let $T_{max}$ represent the maximum time-step. $T_{f,r,c}$ denotes the spike time of the neuron situated at position (r,c) in the feature map *f*, where 0≤f<F, 0≤r<H, 0≤c<W, and $T_{f,r,c}\in \left\{0,1,\ldots ,T_{max}-1\right\}⋃\left\{\infty \right\}$, with symbol *∞* indicating no spike. The input image is transformed from F×H×W dimensions to a four-dimensional spike-wave tensor S[t,f,r,c] of size $T_{max}\times F\times H\times W$. In this tensor, the temporal dimension occupies the first position, and S[t,f,r,c] is computed using Equation (Equation 6).(6)$$S\left[t,f,r,c\right]=\left\{\begin{array}{cc} 0, & t<T_{f,r,c},\\ 1, & \mathrm{otherwise}. \end{array}\right.$$

Figure 2 depicts the process of intensity latency encoding subsequent to the extraction of direction and contrast features using a combination of six filters. Notably, while only five time-step results of intensity latency encoding are displayed in Figure 2, fifteen time-step results are actually utilized in the experiments. It is evident from Figure 2 that neurons associated with higher intensities emit spikes earlier and continue to fire in subsequent time-steps. This cumulative spiking pattern allows for the simultaneous processing of all time-steps and the generation of corresponding outputs, thereby accelerating the operational speed [44].

### 2.2. Architecture Design of ATSNN

Drawing inspiration from traditional CNN architectures, our ATSNN model incorporates a two-layer spiking convolution block, wherein each block comprises an unsupervised spiking convolutional layer followed by a pooling layer. The architectural design of ATSNN is illustrated in Figure 3.

Initially, the original ROI image is fed into a spike encoding layer. Subsequently, the encoded temporal spike signal undergoes processing through two sets of spiking convolutional and pooling layers. Both spiking convolutional layers are constructed using non-leaky integrate-and-fire neurons, which accumulate spikes from presynaptic neurons and emit spikes once their internal membrane potential reaches a predefined threshold. Specifically, each spiking convolutional layer convolves the discrete spike-wave tensor to produce an output membrane potential tensor. Following each spiking convolutional layer, the resulting potential tensor undergoes batch normalization (BN layer) before being utilized to compute the output spike-wave tensor. Here, the adaptive threshold (AT) mechanism plays a crucial role; only neurons whose potentials exceed the firing threshold are permitted to emit spikes. Consequently, both thresholded potential tensors and spike-wave tensors are generated and returned. Subsequently, the output spike-wave tensor is subjected to max pooling for information compression before being passed to the subsequent spiking convolutional layer. Ultimately, the output spike-wave tensors from the final pooling layer are compressed into a one-dimensional tensor by choosing the maximum value along the temporal dimension. This compressed spike-wave tensor represents the final output spike-wave feature representation, which is then utilized for subsequent recognition and classification tasks. For this purpose, we have adopted a straightforward LinearSVC classifier from the sklearn library, configuring it with a specific penalty parameter C=2.4.

#### 2.2.1. Adaptive Threshold Strategy

It is crucial to acknowledge the pivotal role of the firing threshold in neuronal dynamics. Neurons receive inputs from other neurons and determine whether to emit an output spike signal based on this threshold. Typically, a lower threshold increases neuronal susceptibility to activation, whereas a higher threshold necessitates a stronger input signal to elicit neuronal firing. Consequently, by dynamically adjusting the threshold, we can exert control over the activation level and output of neurons, thereby significantly influencing the performance and behavior of neural networks.

To select more appropriate threshold values that better accommodate input data and avoid manual threshold adjustments for varying datasets and tasks, we introduce a novel and straightforward strategy for calculating the adaptive threshold (AT). Specifically, during the training procedure, the AT is dynamically updated using the mean value of the membrane potential tensor after the spiking convolution operation for each image sample. This approach ensures that different images and iterations have distinct firing thresholds, reflecting the high variability of biological neuron firing thresholds under different conditions. Furthermore, the AT setting effectively accommodates various finger vein datasets, enhancing the generalization capability of our ATSNN model across diverse finger vein datasets. The adaptive threshold varies across individual images during the training stage. During the testing stage, the initial adaptive threshold value is set to match the adaptive threshold of the first training image, with subsequent updates based on the iteratively generated adaptive thresholds from the training process, as outlined in Equation (Equation 7):(7)test_threshold=test_threshold×M+train_threshold×(1−M).

Here, *M* represents the momentum parameter, set to 0.9. The momentum parameter in Equation (Equation 7) was evaluated through ablation studies as described in Section 3.5.1. The train_threshold denotes the threshold calculated during the training stage, and test_threshold represents the threshold adopted during the testing stage. The test_threshold is initialized to the adaptive threshold of the first training image and continuously updated with the train_threshold of each image during training.

Additionally, the internal membrane potential of the *i*th neuron is updated at each time step, as defined by Equation (Equation 8):(8)$$V_{i}\left(t\right)=\sum_{j}W_{j,i}S_{j}\left(t-1\right).$$

Specifically, $V_{i}\left(t\right)$ represents the internal potential of the *i*th convolutional neuron at time step *t*, $W_{j,i}$ denotes the synaptic weight between the *j*th presynaptic neuron and the *i*th convolutional neuron, and $S_{j}\left(t-1\right)$ indicates the spike train of the *j*th presynaptic neuron. If $S_{j}\left(t-1\right)$ equals 1, it signifies that the neuron has fired at time step t−1; otherwise, $S_{j}\left(t-1\right)$ equals 0. When $V_{i}$ reaches the firing threshold, the neuron emits a spike.

#### 2.2.2. Local Learning and Weight Update

The convolutional kernel weights are updated based on a simplified STDP learning rule, which is exclusively applied during the training mode. Initially, the synaptic weights of the convolutional neurons are randomly initialized using a normal distribution with a mean of μ=0.8 and a standard deviation of σ=0.05. The weight updates for the two spiking convolutional layers are conducted sequentially; specifically, the weight updates of the first convolutional layer must be completed before initiating the weight updates of the second convolutional layer. During the weight update process for each convolutional layer, the following steps are executed: First, lateral inhibition is applied to the potential tensor obtained after convolution and adaptive thresholding, which eliminates feature redundancy in each spatial region while preserving the most salient features, thereby facilitating the subsequent winner-take-all (WTA) mechanism to identify neurons corresponding to the most distinctive features. Next, a fixed number of winning neurons are selected through the WTA competition strategy for STDP-based weight updates. Finally, a simplified STDP learning rule is employed to determine the input and output spike windows based on the WTA-selected winning neurons and perform the corresponding synaptic weight adjustments.

The lateral inhibition mechanism achieves sparse coding through competitive suppression among feature maps, enforcing that only one neuron is permitted to fire at each spatial location. This ensures that only neurons corresponding to the most salient features are allowed to generate spikes (those with both the earliest firing time and highest membrane potential). This approach effectively reduces redundancy in local input regions, thereby facilitating the WTA mechanism in selecting the most representative feature detectors.The lateral inhibition mechanism of ATSNN was evaluated through ablation studies as described in Section 3.5.3.

The pseudocode for the winner-take-all (WTA) selection of winning neurons is presented in Algorithm 1. The selection of winning neurons is primarily based on the temporal characteristics of spike firing (prioritizing neurons with the earliest firing times), followed by consideration of neuronal membrane potential amplitude (selecting the maximum potential value). To facilitate the learning of distinct features across different feature maps, the algorithm implements a cross-channel inhibition mechanism. Specifically, when a neuron in a given feature map is selected as a winning unit, all neurons within that feature map and those within a predefined spatial neighborhood radius will be inhibited.
**Algorithm 1** Winner-take-all selection mechanism**Require:**$\;\mathrm{potentials}\in R^{T\times F\times H\times W}$: Neuron membrane potentials$\;\mathrm{kwta}\in Z^{+}$: Number of winners inhibition_radius∈N: Lateral inhibition radius spikes: Optional spike tensor**Ensure:** List of winning neurons {(f,h,w)}1:**function** GetKWinners(potentials,kwta,inhibition_radius,spikes)2: **if**
spikes=None
**then**3:   spikes←sign(potentials)          ▹ Binarize potentials4: **end if**5: $\mathrm{first}\_\mathrm{spike}\_\mathrm{time}\leftarrow \mathrm{spikes}.{\mathrm{size}}_{0}-\sum_{t=0}^{T-1}\mathrm{spikes}\left[t\right]$6: first_spike_time←clamp(first_spike_time,0,T−1)7: values←gather(potentials,first_spike_time)8: truncated_pot←spikes⊙values9: v←max(truncated_pot)×T10: truncated_pot←truncated_pot+(spikes×v)11: $\mathrm{total}\leftarrow \sum_{t=0}^{T-1}\mathrm{truncated}\_\mathrm{pot}\left[t\right]$12: winners←∅13: **for**
k←1**to**
kwta
**do**14:  (val,idx)←max(total.reshape(−1))15:  **if**
val=0
**then**16:   **break**17:  **end if**18:  (f,h,w)←unravel_index(idx,total.shape)19:  winners.add((f,h,w))20:  total[f,:,:]←0              ▹ Feature inhibition21:  **if**
inhibition_radius>0
**then**22:   $h_{\min}\leftarrow \max\left(0,h-r\right)$23:   $h_{\max}\leftarrow \min\left(H,h+r+1\right)$24:   $w_{\min}\leftarrow \max\left(0,w-r\right)$25:   $w_{\max}\leftarrow \min\left(W,w+r+1\right)$26:   $\mathrm{total}[:,h_{\min}:h_{\max},w_{\min}:w_{\max}]\leftarrow 0$27:  **end if**28: **end for**29: **return**
winners30:**end function**

Based on the winners selected by WTA, STDP can precisely locate the windows of presynaptic and postsynaptic spikes in the input and output spike feature maps that participate in weight updates, thereby achieving synaptic weight regulation. Here, the input spike feature maps correspond to the spike inputs to each convolutional layer, while the output spike feature maps are generated by applying adaptive thresholding to the potential tensors obtained generated through the convolution operation between the convolutional kernel and the input spike feature map. STDP is a biological mechanism that regulates the strength of synaptic connections between neurons. If presynaptic neurons emit action potentials before postsynaptic neurons, it is termed pre-post timing, resulting in an increase in synaptic weights to strengthen synaptic connections. Conversely, if a presynaptic neuron fires after a postsynaptic neuron emits an action potential, it is termed post-pre timing, leading to a decrease in synaptic weight to weaken the synaptic connection. For simplicity, the STDP learning rule adopted in ATSNN follows the definition outlined in [44], and its update equations are defined as follows:(9)$$\Delta W_{i,j}=\left\{\begin{array}{cc} A^{+}\times \left(W_{i,j}-\mathrm{LB}\right)\times \left(\mathrm{UB}-W_{i,j}\right) & \mathrm{if}\;T_{j}\leq T_{i},\\ A^{-}\times \left(W_{i,j}-\mathrm{LB}\right)\times \left(\mathrm{UB}-W_{i,j}\right) & \mathrm{if}\;T_{j}>T_{i}. \end{array}\right.$$
where $\Delta W_{i,j}$ represents the amount of weight change for the synapse connecting postsynaptic neuron *i* and presynaptic neuron *j*, and $A^{+}$ and $A^{-}$ are the learning rates corresponding to potentiation and depression levels, respectively, with $A^{+}>0$ and $A^{-}<0$. The term $\left(W_{i,j}-\mathrm{LB}\right)\times \left(\mathrm{UB}-W_{i,j}\right)$ serves as a stabilizer that slows down weight changes when the synaptic weight $W_{i,j}$ approaches the lower limit LB or the upper limit UB. Notably, the synaptic weight change does not account for the exact time difference between two spikes but only considers the order of the spikes.

The pseudocode for STDP-based weight updates is presented in Algorithm 2. The mechanism operates by first identifying the windows of presynaptic and postsynaptic spikes involved in synaptic plasticity based on each winner, followed by adjusting synaptic weights according to the order of these presynaptic and postsynaptic firing events.
**Algorithm 2** STDP learning rule**Require:**$\mathrm{input}\_\mathrm{spikes}\in {\left\{0,1\right\}}^{T\times C_{\mathrm{in}}\times H_{\mathrm{in}}\times W_{\mathrm{in}}}$: Presynaptic spikes$\mathrm{potentials}\in R^{T\times F\times H\times W}$: Postsynaptic potentialsoutput_spikes: Postsynaptic spikeswinners: Optional precomputed winners1:**procedure** STDPUpdate(input_spikes,potentials,output_spikes,winners)2:  **if**
winners=None
**then**3:   winners←GetKWinners(potentials,kwta,inhibition_radius,output_spikes)4:  **end if**5:  pairings←∅6:  **for**
(f,h,w)∈winners
**do**7:   $\tau_{\mathrm{post}}\leftarrow \sum_{t=0}^{T-1}\mathrm{output}\_\mathrm{spikes}\left[t,f,h,w\right]$8:   $\mathrm{post}\_\mathrm{window}\leftarrow \mathrm{ones}\left(\mathrm{kernel}\_\mathrm{size}\right)\times \tau_{\mathrm{post}}$9:   pre_window←input_spikes[:,:,h−r:h+r,w−r:w+r]10:   pairings.add(pre_window≥post_window)11:  **end for**12:  $\Delta W\leftarrow 0_{C_{\mathrm{in}}\times F\times k_{h}\times k_{w}}$13:  **for**
i∈{0,…,|winners|−1}
**do**14:   $f\leftarrow \mathrm{winners}{\left[i\right]}_{0}$15:   $\Delta W\left[:,f,:,:\right]\leftarrow \mathrm{where}(\mathrm{pairings}\left[i\right],A^{+},A^{-})$16:  **end for**17:  **if**
use_stabilizer
**then**18:   ΔW←ΔW⊙(W−LB)⊙(UB−W)19:  **end if**20:  W←W+η⊙ΔW            ▹η: learning rate21:  W←clamp(W,LB,UB)22:**end procedure**

The STDP algorithm serves as the weight update mechanism in the ATSNN model, employing a winner neuron-driven local connection update strategy that achieves dual optimization while maintaining network sparsity. Compared to the fully-connected weight update approach in traditional convolutional neural networks, the STDP mechanism significantly reduces the number of trainable parameters, thereby markedly improving computational efficiency. Furthermore, the STDP mechanism enables the network to adaptively enhance the most discriminative high-frequency feature representations in input images through dynamic synaptic weight adjustments, effectively boosting the network’s feature extraction capability. This biologically inspired learning mechanism not only reduces computational resource consumption but, more importantly, simulates the plasticity dynamics of neural systems, allowing the model to autonomously discover and strengthen the most representative feature patterns in input data.

### 2.3. Classification and Recognition

After the final pooling layer, the spike-wave tensor undergoes a two-step preprocessing procedure before being fed into a support vector machine (SVM) for classification. Initially, the maximum value along the temporal dimension is extracted to reduce dimensionality. Subsequently, this processed tensor is further compressed into a one-dimensional tensor, suitable for input into the LinearSVC classification function from the Scikit-learn library. LinearSVC, which is grounded in the linear SVM algorithm, seeks to delineate an optimal hyperplane within the feature space to discern between various classes of samples. In the context of finger vein recognition tasks, LinearSVC adopts a one-vs-rest strategy during the training phase. This involves constructing multiple binary classifiers, each tasked with distinguishing a single class from the remaining classes. During the testing phase, a novel test sample is sequentially input into each of these binary classifiers for prediction. Ultimately, the class label associated with the highest confidence score (predicted value) is chosen as the final predicted result. The detailed steps are outlined below:

For each category *i*:Label samples belonging to category *i* as the positive class (1) and samples from other categories as the negative category (0).Utilize the labeled dataset to train a binary classifier.Obtain a classifier specific to category *i*.

For a new prediction sample *x*:For each classifier corresponding to category *i*, compute the confidence score (predicted value) indicating the likelihood that sample *x* belongs to category *i*.Select the category with the highest confidence score as the prediction outcome.

## 3. Experimental Results and Discussion

This section is structured as follows. Section 3.1 provides a concise overview of the FV datasets employed in our experiments. Specifically, we utilize three benchmark FV databases: MMCBNU_6000 [17], FV-USM [46], and SDUMLA-HMT [47]. Section 3.2 details the relevant parameter settings of the ATSNN model used in our study. Section 3.3 outlines the evaluation metrics employed to assess the performance of our model. Section 3.4 delves into the sensitivity analysis of the ATSNN model, focusing specifically on the impact of various parameter settings. This includes an examination of the size and type of filters, the resizing of the input ROI image, and the number of extracted features. Section 3.5 presents ablation studies that systematically investigate three key components: First, the selection of momentum parameter values in the adaptive threshold mechanism is examined. Subsequently, the influence of the adaptive threshold and batch normalization layers on model performance is analyzed. Finally, the impact of lateral inhibition mechanisms on the model is explored.

Finally, Section 3.6 compares our ATSNN model with several mainstream deep learning-based FV recognition methods, providing a comprehensive evaluation of its effectiveness.

### 3.1. Finger Vein Datasets

MMCBNU_6000 is a publicly accessible finger vein dataset comprising finger vein image samples from 100 volunteers spanning 20 countries. Each volunteer’s six fingers were captured 10 times, yielding a total of 60 finger vein images per individual and an aggregate of 6000 images. These images are stored in BMP format with a resolution of 480×640 pixels. The dataset includes ROI images, which have had the finger contours removed, retaining only the vein region, resulting in a size of 60×128 pixels. These ROI images are directly used in our experiments.

The FV-USM database, collected by University Sains Malaysia, contains finger vein images from 123 volunteers. Data collection was conducted in two stages, with each stage capturing four fingers from each volunteer, with each finger being imaged six times. Consequently, there are 2952 images per stage, each with a resolution of 640×480 pixels. The ROI images for the first and second stages measure 150×50 and 300×100 pixels, respectively.

The SDUMLA database, provided by Shandong University of China, encompasses finger vein images from 106 volunteers. Each volunteer’s six fingers were captured six times, resulting in a total of 36 images per individual and a dataset comprising 3816 finger vein images in total. These images are stored in BMP format with dimensions of 240×320 pixels.

Unlike the MMCBNU_6000 and FV-USM datasets, the SDUMLA-HMT dataset does not provide pre-extracted ROI images. To obtain ROI images from the SDUMLA-HMT dataset, we performed an extraction process. This involved cropping 45 pixels from the top and 25 pixels from the bottom of each image to minimize the impact of device edges on contour extraction. Subsequently, Canny edge detection was employed to locate the upper and lower contours of the ROI, with unnecessary vertical device contours being removed. Finally, the ROI images were cropped based on these contours and resized uniformly using bilinear interpolation to a specified resolution. The extraction process is illustrated in Figure 4.

For comparison, Figure 5 presents original and ROI image samples selected from the three datasets, while Table 1 outlines the relevant attributes of these datasets.

### 3.2. Experimental Parameters Setting

In order to expedite the training process and enhance the accuracy of finger vein recognition, ROI images from the finger vein datasets were used in the experiment without any prior preprocessing. Specifically, for the MMCBNU_6000 dataset, the first nine images of each finger were designated as the training set, whereas the remaining one image served as the test set, yielding a total of 5400 training images and 600 test images. For the FV-USM dataset, five images from each finger class within each stage were selected for the training set, with the remaining one image allocated to the test set, resulting in a total of 4920 training images and 984 test images. In the case of the SDUMLA-HMT dataset, one image from each finger class was chosen as the test set, while the remaining five images were utilized for training, leading to a total of 3180 training images and 636 test images. It is noteworthy that the ROI images of the FV-USM dataset were rotated counterclockwise by $90^{\circ }$ to align with the finger orientations of the other datasets.

The network architecture of the ATSNN is depicted in Figure 3. It comprises a spike encoding layer, two AT spike convolutional layers, and two pooling layers, followed by classification using the LinearSVC classifier. The spike encoding layer employs four Gabor filters and two DoG filters. The parameters of the four Gabor kernel are set as follows: a window size of 15×15, with orientations of 3π/8, 5π/8, 7π/8 and 9π/8. The two DoG kernels have initial window sizes of 7×7, with standard deviations of 1 and 2 for the first kernel, and 2 and 1 for the second kernel. Intensity latency encoding is conducted using 15 time steps. The first spiking convolutional layer consists of 16 feature maps, with a convolution window size of 5×5, using AT with five winners and a lateral inhibition radius of 2. The second spiking convolutional layer comprises 20 feature maps, with a convolution window size of 2×2, employing AT with 8 winners and a lateral inhibition radius of 1. The pooling layers have a pooling window size of 2×2, a stride of 2, and padding of 1. The detailed parameter settings for the convolutional and pooling layers are outlined in Table 2.

The synaptic weights of the convolutional neurons are initialized using random values sampled from a normal distribution with a mean (μ) of 0.8 and a standard deviation (σ) of 0.05. The learning rates for all convolutional layers, based on the STDP learning rule, are set to $A^{+}=0.004$ and $A^{-}=-0.003$. To expedite convergence, a learning rate adjustment mechanism is incorporated into the first convolutional layer. Specifically, whenever the learning rate drops below 0.15, the learning rate $A^{+}$ is doubled every 500 training images, while maintaining the initial ratio between $A^{+}$ and $A^{-}$. The penalty parameter for the LinearSVC classifier is set to C=2.4.

The hierarchical learning strategy of the ATSNN enables the network to progressively construct robust feature representations. The primary task of the first layer is to extract shallow features, such as edges, textures, and corners, which represent low-level visual information. These features are relatively simple and easy to learn, and typically, two epochs of training are sufficient for the network to capture these fundamental patterns. In contrast, the second layer integrates these details into higher-level discriminative features, learning more complex and abstract semantic information, such as object shapes, structures, and higher-order patterns. These advanced features require more training time to gradually optimize and integrate. Generally, 20 epochs of training provide the network with adequate time to explore the diversity of the data and learn more discriminative features. Therefore, the training epochs for the first convolutional layer are set to 2, while the second convolutional layer is trained for 20 epochs. This design ensures that the network efficiently extracts low-level features while thoroughly optimizing high-level representations, ultimately enhancing the model’s performance and generalization capability.

Finally, it is worth noting that all experiments were conducted using Python 3.9 with the PyTorch 1.10.2 framework, on a desktop PC equipped with an Intel Core i5 CPU operating at 3.7 GHz, 16 GB of RAM (Intel, Santa Clara, CA, USA), and an NVIDIA GeForce RTX 2080 SUPER GPU (Nvidia, Santa Clara, CA, USA).

### 3.3. Evaluation Metrics

To quantitatively evaluate the performance of the ATSNN model, we employed several standard metrics in our experiments.

*Accuracy*, which is defined as the ratio of correctly classified sample pairs to the total number of sample pairs, as given by Equation (Equation 10), where $N_{T}$ represents the number of correctly classified sample pairs and *N* represents the total number of sample pairs.(10)$$\mathrm{Accuracy}=\frac{N_{T}}{N}\times 100\%.$$*FAR* (false acceptance rate), which measures the proportion of false acceptance cases among all heterogeneous matching cases, as calculated in Equation (Equation 11). Here, FA represents the number of false acceptance cases, and TA represents the number of true acceptance cases. In classification problems, a false acceptance case occurs when two samples of different types are mistakenly recognized as being of the same type. Therefore, FAR can be interpreted as a metric of inter-class distance, with a lower FAR indicating better inter-class separation.(11)$$\mathrm{FAR}=\frac{FA}{FA+TA}.$$*FRR* (false rejection rate), which quantifies the proportion of false rejection cases among all matching cases of the same type, as shown in Equation (Equation 12). In this context, FR represents the number of false rejection cases, and TR represents the number of true rejection cases. A false rejection case arises when two samples of the same type are mistakenly recognized as being dissimilar by the system. Hence, FRR serves as a metric to measure intra-class correlation, with a lower FRR indicating better intra-class similarity.(12)$$\mathrm{FRR}=\frac{FR}{FR+TR}.$$*EER* (equal error rate), which is defined as the ratio of trials in which the FAR is equal to the FRR. A lower EER indicates better performance in the FV verification tasks.

### 3.4. Analysis on the Parameters Sensitivity

In this section, we undertake a comprehensive analyze of the sensitivity of key parameters within the ATSNN model. Specifically, we examine the impact of the size and type of image spike encoding filters, the dimensions of input ROI images, and the number of features utilized for final classification. All experimental evaluations are conducted using three FV datasets, with a rigorous approach ensuring that when assessing a particular parameter, the remaining parameters are held constant at the values specified in Section 3.2.

The ATSNN, as proposed, comprises a two-layer network architecture. Consequently, the dimensions of the input image and the size of the filtering kernel employed during spike encoding have a profound influence on the recognition accuracy of the ATSNN. In shallow neural networks, individual neurons typically possess a smaller receptive field, enabling them to perceive only local information within the input image. Conversely, deep networks are characterized by larger receptive fields, which are progressively expanded through the stacking of multiple convolutional and pooling layers. This hierarchical structure allows deep networks to effectively handle larger images and capture more global features. However, for shallow neural networks such as ATSNN, identifying appropriate image and filtering kernel sizes is crucial. These dimensions must be such that the receptive field of the ATSNN can maximize its coverage of global information within the images. By doing so, we aim to achieve superior recognition accuracy with more compact network models, thereby reducing training time. Furthermore, the deployment of finger vein recognition tasks necessitates models of smaller size and faster recognition speeds. In light of these considerations, subsequent experiments will focus on exploring and identifying the most suitable image and filtering kernel sizes for shallow ATSNN architectures.

#### 3.4.1. Size and Type of Filter

During the spike encoding stage of finger vein image processing, the dimensions and type of the filtering kernel significantly impact the extraction of finger vein features, ultimately determining the quality of spike encoding. To investigate this influence, we conducted an analysis using three FV datasets, with the ROI images from the SDUMLA-HMT dataset resized to 60×160 pixels to maintain consistency with the other datasets. Figure 6 illustrates an ROI image from the MMCBNU_6000 dataset, filtered using Gabor and DoG kernels of varying scales and orientations. Specifically, each scale of the Gabor filter bank comprises kernels oriented at 3π/8, 5π/8, 7π/8 and 9π/8 radians, whereas each scale of the DoG filter bank includes kernels with pairs of Gaussian standard deviations (δ) of 1 and 2, and 2 and 1, respectively. From Figure 6, it is evident that Gabor kernels of different scales exhibit varying capabilities in extracting orientation information from finger vein images. Notably, the off-center kernel ($\delta_{1}=2$ and $\delta_{2}=1$) within the DoG set captures clearer contrast information of finger veins, attributed to the higher pixel intensity in the finger vein regions compared to the background.

Table 3, Table 4 and Table 5 present the quantitative impact of various filtering kernel sizes on finger vein recognition performance, encompassing classification accuracy, training time, and EER across the three datasets. The experimental results demonstrate that the combined use of Gabor and DoG kernels for finger vein image encoding enhances classification performance in the ATSNN. This is attributed to the fact that Gabor kernels are adept at extracting orientational information from finger vein images, while DoG kernels effectively capture contrast information. The employment of multiple filters diversifies the spike encoding, contributing to improved recognition outcomes.

#### 3.4.2. Size of ROI Image

The size of the input finger vein ROI images can exert a considerable influence on the recognition accuracy of the ATSNN. Specifically, for small-sized images, the receptive field of ATSNN can adequately cover essential local information within the image, potentially leading to robust performance. Conversely, for large-sized images, the receptive field of ATSNN may fail to capture global information adequately, potentially resulting in a decline in recognition performance. Additionally, reducing the image size has the benefit of decreasing both training and classification time. Therefore, we conducted an examination of the impact of varying ROI image sizes on the recognition performance of ATSNN.

Table 6 presents the comprehensive results of this investigation, detailing the effects of different finger vein ROI image sizes on recognition accuracy, training time, and EER of ATSNN across three finger vein datasets. Notably, the optimal sizes of filtering kernels vary depending on the image sizes, with smaller images often necessitating smaller kernel sizes. The results indicate that further reduction in image size not only decreases both training and classification time but, in the case of the FV-USM dataset, also leads to an enhancement in recognition accuracy.

#### 3.4.3. Feature Selection

In this section, we aim to further decrease classification time while observing the corresponding changes in accuracy through the process of feature selection. Feature selection is a technique aimed at identifying the optimal subset of features by eliminating those that are irrelevant or redundant. This reduction in the number of features can enhance model accuracy, expedite runtime, and mitigate computational costs. Given that feature extraction in the ATSNN is decoupled from the classification process, we adopt the SelectPercentile method of filter-based feature selection. The SelectPercentile function, a feature selection function within the Scikit-learn library, evaluates and selects features based on the chi-square test between the features and the target variable, thereby filtering out those that exhibit significant correlations with the classification objective. Subsequently, we utilize the LinearSVC classification function from the Scikit-learn library for classification purposes.

Table 7 presents the comprehensive results of our investigation into the impact of varying feature percentages in feature selection on recognition accuracy, training time, classification time, and EER. The results indicate that further application of feature selection can result in reduced classification time while maintaining high accuracy and a low EER. In scenarios where a faster recognition speed is prioritized, a slight compromise in accuracy can lead to even quicker recognition times.

### 3.5. Ablation Study

In this subsection, we conducted an ablation study to evaluate the significance of key components within the ATSNN model. Specifically, we first investigated the configuration of the momentum parameter in the adaptive threshold (AT) mechanism, followed by a comparative analysis of network performance using AT versus fixed threshold, and then examined the impact of incorporating batch normalization (BN) layers after each spiking convolutional layer. Finally, we explored the effects of the lateral inhibition mechanism on model behavior. It is important to note that when analyzing one type of structure, the other structures and parameter settings were held constant, adhering to the configurations reported in Section 3.2.

#### 3.5.1. Momentum Parameter

Table 8 presents the recognition accuracy, EER value, and model training and classification time achieved with different momentum parameters in the adaptive threshold on the MMCBNU_6000 dataset. All ROI images in the MMCBNU_6000 dataset were resized to 60×128 dimensions. In adaptive thresholding algorithms, the momentum parameter serves as a critical hyperparameter whose value significantly impacts model performance. Experimental results demonstrate that when the momentum parameter is set to 0.9, the model achieves optimal comprehensive performance on the MMCBNU dataset, attaining both the highest recognition accuracy and the lowest equal error rate (EER). This finding indicates that a momentum value of 0.9 effectively maintains threshold update stability while simultaneously adapting to variations in data characteristics, thus being selected as the final parameter configuration.

#### 3.5.2. Adaptive Threshold and Batch Normalization Layers

Table 9 presents the recognition accuracy, EER value, and model training and classification time achieved using various threshold strategies, as well as the addition of BN layers following each spiking convolutional layer on the FV-USM dataset. All ROI images in the FV-USM dataset were resized to 48×48 dimensions. In Table 9, Convt_1 and Convt_2 denote the fixed thresholds applied to the first and second spiking convolutional layers, respectively. AT indicates the usage of adaptive thresholds for both spiking convolutional layers, while AT+BN signifies the employment of adaptive thresholds for both layers supplemented by a BN layer after each spiking convolutional layer.

The experimental results revealed that when using fixed thresholds, a smaller fixed threshold for the first convolutional layer resulted in a lower EER. This observation can be attributed to the smaller image size utilized in the experiments. Smaller images inherently contain less information, and reducing the fixed threshold for the first layer facilitates the transmission of more image information as spike signals to subsequent layer, thereby enabling the extraction of more features for classification. Conversely, the integration of adaptive thresholds and the addition of BN layers after each spiking convolutional layer achieved the lowest EER value. This outcome is attributed to the ability of adaptive thresholds to adapt to diverse images and training processes, mimicking the highly variable firing thresholds of biological neurons under different conditions. Furthermore, the incorporation of BN layers enhanced the training stability of the model, ultimately contributing to superior performance.

#### 3.5.3. Lateral Inhibition Mechanisms

Table 10 presents the effects of lateral inhibition mechanisms on model performance metrics, including recognition accuracy, EER value, and model training and classification time evaluated on the MMCBNU_6000 dataset. All ROI images in the MMCBNU_6000 dataset were resized to 60×128 dimensions. The ATSNN model implements lateral inhibition mechanisms at three distinct stages during network training: potential inhibition applied to thresholded post-convolution potentials in both the first and second convolutional layers to optimize winner selection in the WTA mechanism, along with spike inhibition applied to inputs of the second convolutional layer. The potential inhibition preceding WTA operation serves to minimize regional redundancy within feature maps, thereby enhancing the WTA mechanism’s capacity to identify neurons corresponding to the most salient features. Concurrently, the inter-layer spike inhibition induces sparsity in cross-channel responses at identical spatial coordinates, driving the network to establish orthogonal feature representations across different channels and consequently augmenting the diversity of learned feature representations.

Experimental results demonstrate that removing lateral inhibition either before the WTA mechanism in either the first or second convolutional layer reduces model accuracy. Although eliminating lateral inhibition of spike inputs to the second convolutional layer maintains accuracy levels, it increases training and classification time. These findings further confirm that lateral inhibition serves two crucial roles. First, when applied before WTA operations, it enhances the WTA mechanism’s selection of the most salient neurons, thereby improving subsequent STDP-based weight updates. Second, when implemented for spike inputs between convolutional layers, it enhances feature discriminability, thereby reducing training and classification time.

### 3.6. Comparison Experiments

In this experiment, we compared the proposed ATSNN with several prominent deep learning-based models that have demonstrated success in the face verification domain, including AlexNet [23], ResNet [52], VGG [53], DenseNet-161 [24], and a lightweight CNN enhanced with CBAM [31]. To ensure consistency across all models, we maintained uniform dataset partitioning and did not utilize pre-trained network models.

To accommodate mainstream deep learning architectures, we adjusted the sizes of the ROI images for the MMCBNU_6000, FV-USM, and SDUMLA-HMT datasets. Specifically, the ROI image sizes were set to 120×256 for MMCBNU_6000, 300×100 for FV-USM, and 120×320 for SDUMLA-HMT. However, given that the study in [31] employed a two-layer network and demonstrated its effectiveness with smaller-sized images, we adjusted the ROI image sizes accordingly for the replication of [31]: 60×128 for MMCBNU_6000, 150×50 for FV-USM, and 60×160 for SDUMLA-HMT. This adjustment ensured that the receptive field size adequately covered crucial local information in the images while preventing excessive loss of image information that could adversely impact classification performance.

Table 11, Table 12, and Table 13 present the recognition accuracy, EER, model parameter quantities, and FLOPs obtained by the compared models on the MMCBNU_6000, FV-USM, and SDUMLA-HMT datasets, respectively. Our ATSNN achieved the best accuracy and EER on all three FV datasets, with the smallest model parameters and FLOPs. Specifically, for the MMCBNU_6000 dataset, ResNet18, VGG16, and DenseNet161 followed closely in terms of accuracy and EER, but their model parameters and FLOPs were significantly greater than those of the ATSNN. For the FV-USM dataset, VGG16 and DenseNet161 ranked second in accuracy and EER, while for the SDUMLA-HMT dataset, ResNet34, VGG16, DenseNet161, and the CNN+CBAM model occupied the second position. Notably, our ATSNN consistently delivered the lowest EER across all datasets, accompanied by the smallest model parameters and FLOPs. Furthermore, our comparison between ResNet18, ResNet34, and ResNet50 revealed that, on a finger vein dataset with limited samples per class, the non-pretrained ResNet18 and ResNet34 models outperformed ResNet50. This suggests that shallower models such as ResNet18 and ResNet34 are more adept at generalizing to smaller training datasets, leading to superior performance. This finding underscores the advantage of using shallow networks for finger vein recognition, as they not only reduce the number of parameters and computational requirements but also demonstrate that non-pretrained shallow networks are more prone to achieving excellent recognition performance on small-sample datasets.

Ultimately, to evaluate the efficacy of our ATSNN model against state-of-the-art finger vein verification methods, we selected four recently published hand-crafted methods: sparse reconstruction error constrained low-rank representation (SRLRR) [54], local phase quantization (LPQ) [55], Radon-like features (RLFs) [56], and partial least squares discriminant analysis (PLS-DA) [57]. Additionally, we compared our model against seven deep learning-based models: the Rectangular Filter Finger-Vein Recognition Network (Rec-FFVN) and Semi Pre-trained Finger-Vein Recognition Network (Semi-PFVN) [58], a two-stream CNN [29], a fully convolutional network (FCN) [59], a CNN competitive order (CNN-CO) [60], a convolutional autoencoder (CAE) [61], and a lightweight CNN combining center loss and dynamic regularization (Lightweight CNN) [62].

As illustrated in Table 14, deep learning-based models generally exhibit lower EERs compared to hand-crafted feature-extraction methods. This superiority stems primarily from the fact that hand-crafted shallow features are prone to noise, image rotation, and translation, and lack robust expressive power. Conversely, deep learning models automatically derive higher-level features with enhanced generalization capabilities. Notably, our proposed ATSNN model attains the lowest EER across all three datasets among the compared models. This outperformance can be attributed to the intensity latency encoding scheme tailored for finger vein images and the novel adaptive threshold for spike firing. The ATSNN model not only demonstrates exceptional performance but also leverages spike-based information transmission and feature extraction, inherently possessing lightweight characteristics. Furthermore, our method obviates the need for additional preprocessing of finger vein images, thereby substantially reducing computational time.

Experimental results demonstrate that the ATSNN model achieves superior performance across three finger vein datasets while maintaining significantly fewer parameters and FLOPs compared to conventional models. The model’s advantages manifest in two key aspects: computational efficiency and feature learning capability. In terms of computational efficiency, the event-driven nature of spiking neural networks combined with local connection update strategies substantially reduces computational resource requirements. Regarding feature learning, the STDP mechanism emulates spike-timing-dependent plasticity in biological neural systems, enabling autonomous discovery and reinforcement of the most discriminative feature representations from input data. This biologically inspired design not only delivers outstanding recognition accuracy but also achieves remarkable reductions in model complexity and computational overhead. Furthermore, the lightweight architecture makes the model particularly suitable for deployment on resource-constrained embedded devices, offering a viable technical solution for developing practical finger vein recognition systems.

## 4. Conclusions and Future Research

In this paper, we introduce an ATSNN model for finger vein recognition. Firstly, the model determines neuron firing using an adaptive threshold mechanism, which aligns with the highly variable threshold behavior of biological neurons and obviates the need for manual threshold adjustment. This enhancement bolsters the network’s generalization capability. Secondly, the integration of a batch normalization layer with the adaptive threshold mechanism further improves the network’s stability during training. Additionally, the model employs multiple filters to extract directional and contrast information from finger vein images, thereby enriching the spike encoding process. ATSNN demonstrates superior performance compared to state-of-the-art finger vein verification models across three benchmark FV datasets—MMCBNU_6000, FV-USM, and SDUMLA-HMT—achievinh EERs of 0.08%, 0.10%, and 0.08%, respectively. Furthermore, the proposed ATSNN model boasts minimal parameters and FLOPs due to its streamlined architecture, comprising only two spiking convolutional layers, utilizing spike-based information transmission between layers, and employing an SVM classifier for classification based on extracted spike features. Notably, our approach forgoes pre-training, allowing network parameters to be initialized using a normal distribution. Moreover, our method eliminates the need for extra preprocessing of finger vein images, significantly reducing time costs. Consequently, ATSNN exhibits a low overall training cost, rendering it suitable for deployment in computationally constrained environments.

While the proposed ATSNN model has achieved remarkable performance in finger vein recognition, it is important to note that the feature extraction process, based on unsupervised SNN, does not leverage label information for feedback regulation. This may lead to the extraction of redundant features for the SVM classifier. In future work, we aim to explore the incorporation of label information to extract more diagnostically meaningful features, thereby further refining the performance of our model.

## Figures and Tables

**Figure 1 sensors-25-02279-f001:**
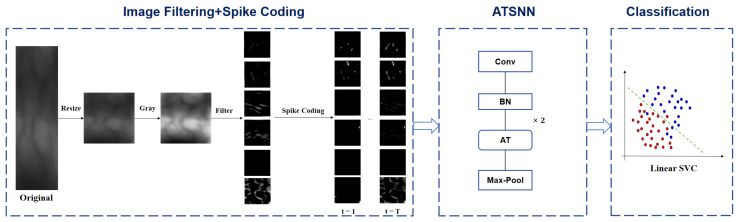
Schematic diagram of ATSNN-based finger vein recognition workflow.

**Figure 2 sensors-25-02279-f002:**
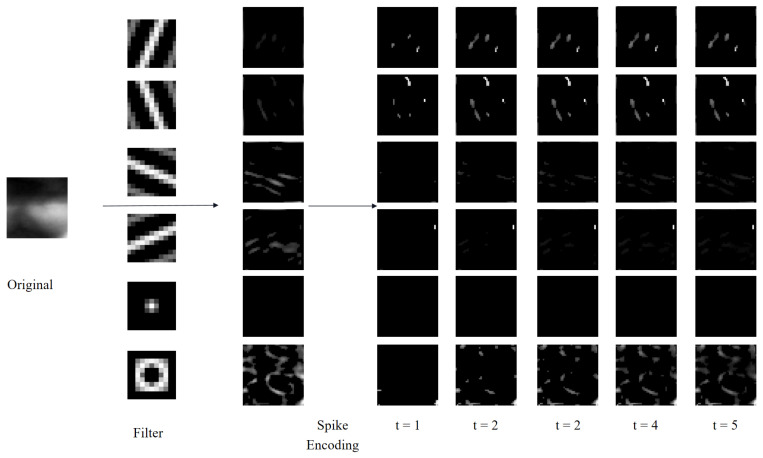
Image filtering and intensity delay encoding.

**Figure 3 sensors-25-02279-f003:**
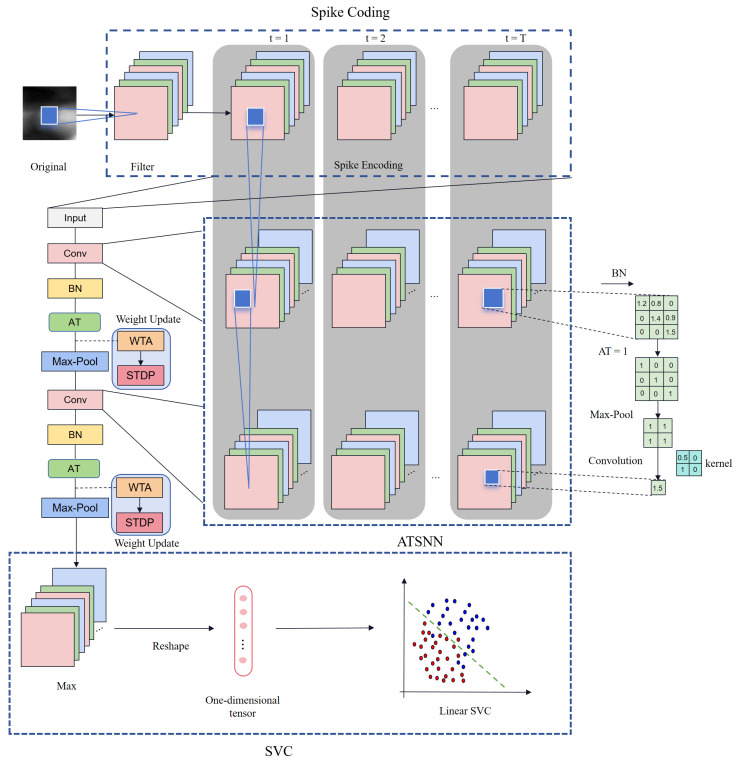
Architecture design of the adaptive threshold spiking convolutional neural network (ATSNN).

**Figure 4 sensors-25-02279-f004:**
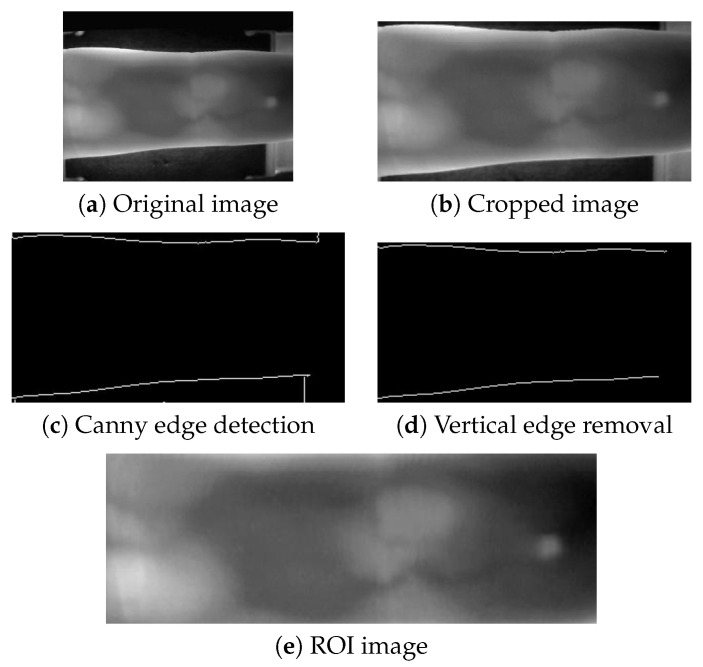
ROI image capture process of SDUMLA-HMT.

**Figure 5 sensors-25-02279-f005:**
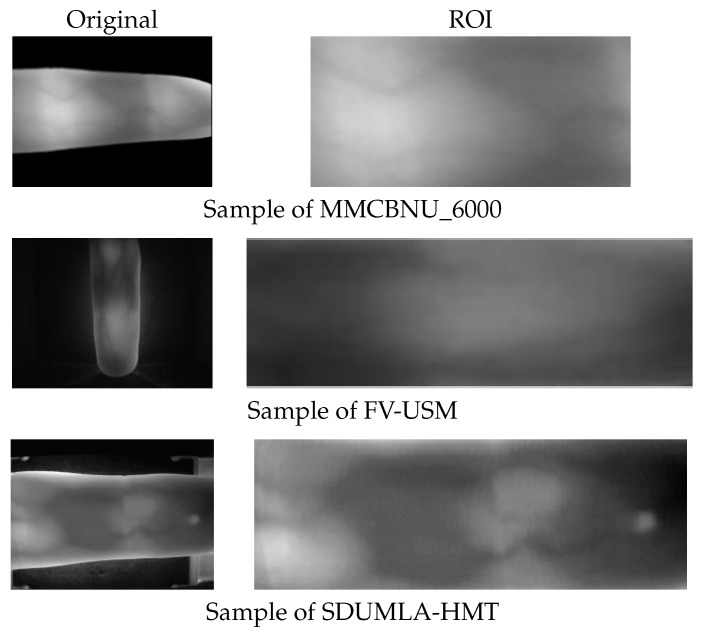
Original and ROI sample images of three FV datasets.

**Figure 6 sensors-25-02279-f006:**
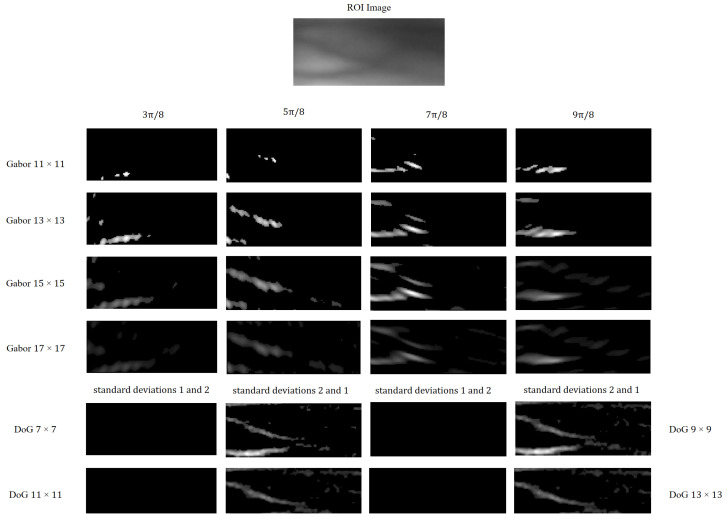
The filtered results of Gabor kernels and DoG kernels under different scales and orientations.

**Table 1 sensors-25-02279-t001:** Details of three finger vein databases.

Databases	Num of Individuals	Fingers	Hands	Num of Images per Finger	Sessions	Num of Finger Classes	Total Num of Images	Image Size	ROI Image Size
MMCBNU_6000	100	index, middle, ring	left, right	10	1	600	6000	480×640	60×128
FV-USM	123	index, middle	left, right	12	2	492	5904	640×480	150×50,300×100
SDUMLA-HMT	106	index, middle, ring	left, right	6	1	636	3816	240×320	120×320

**Table 2 sensors-25-02279-t002:** Parameters setting of the spiking convolutional layer and pooling layer in the ATSNN.

Layer	Number of Feature Maps	Input Window (Width, Height, Depth)	Stride	Threshold
Conv-1	16	5×5×6	1	adaptive threshold
Pooling-1	16	2×2	2	-
Conv-2	20	2×2×16	1	adaptive threshold
Pooling-2	20	2×2	2	-

**Table 3 sensors-25-02279-t003:** The recognition accuracy, EER value, and training and classification time of the ATSNN model under different kernel sizes on the MMCBNU_6000 dataset.

Kernel Type	Kernel Size	*Accuracy* (%)	*EER* (%)	Time (s)
DoG	7×7	99.50	0.25	630
DoG	9×9	99.16	0.42	436
DoG	11×11	99.50	0.25	576
DoG	13×13	99.16	0.42	503
Gabor	11×11	95.17	2.42	786
Gabor	13×13	98.83	0.58	638
Gabor	15×15	99.50	0.25	462
Gabor	17×17	99.16	0.42	374
Gabor + DoG	15×15, 7×7	**99.83**	**0.08**	459
Gabor + DoG	15×15, 9×9	99.66	0.17	435
Gabor + DoG	15×15, 11×11	99.28	0.08	459

**Table 4 sensors-25-02279-t004:** The recognition accuracy, EER value, and training and classification time of the ATSNN model under different kernel sizes on the FV-USM dataset.

Kernel Type	Kernel Size	Accuracy (%)	EER (%)	Time (s)
DoG	7×7	97.96	1.02	590
DoG	9×9	98.06	0.97	692
DoG	11×11	98.27	0.87	576
DoG	13×13	96.84	1.58	493
Gabor	11×11	70.42	14.82	1039
Gabor	13×13	96.23	1.88	697
Gabor	15×15	99.49	0.25	593
Gabor	17×17	99.08	0.46	450
Gabor + DoG	15×15, 7×7	99.28	0.36	603
Gabor + DoG	15×15, 9×9	99.39	0.31	505
Gabor + DoG	15×15, 11×11	**99.59**	**0.20**	612

**Table 5 sensors-25-02279-t005:** The recognition accuracy, EER value, and training and classification time of the ATSNN model under different kernel sizes on the SDUMLA-HMT dataset.

Kernel Type	Kernel Size	Accuracy (%)	EER (%)	Time (s)
DoG	7×7	99.52	0.24	424
DoG	9×9	99.52	0.24	448
DoG	11×11	99.52	0.24	387
DoG	13×13	99.37	0.31	336
Gabor	11×11	99.52	0.24	523
Gabor	13×13	99.52	0.24	470
Gabor	15×15	99.68	0.16	401
Gabor	17×17	99.68	0.16	230
Gabor + DoG	15×15, 7×7	**99.84**	**0.08**	427
Gabor + DoG	15×15, 9×9	99.68	0.16	348
Gabor + DoG	15×15, 11×11	99.68	0.16	333

**Table 6 sensors-25-02279-t006:** The recognition accuracy, EER value, and model training and classification time of the proposed method on the ROI image of different sizes of the three datasets.

Databases	ROI Image Size	Accuracy (%)	EER (%)	Time (s)
MMCBNU_6000	** 60×128 **	**99.83**	**0.08**	459
	64×64	99.67	0.17	452
	30×64	99.67	0.17	387
	48×48	99.67	0.17	372
FV-USM	50×150	99.59	0.20	612
	64×64	99.59	0.20	379
	25×75	99.59	0.20	301
	** 48×48 **	**99.79**	**0.10**	350
SDUMLA-HMT	60×160	99.84	0.08	427
	64×64	99.84	0.08	292
	30×80	99.68	0.16	208
	** 48×48 **	**99.84**	**0.08**	208

**Table 7 sensors-25-02279-t007:** The recognition accuracy, EER value, and classification time of different characteristic percentage on the ROI image of the three datasets.

Databases	Characteristic Percentage (%)	Accuracy (%)	EER (%)	Time (s)
MMCBNU_6000	20	99.50	0.25	23.66
	50	99.50	0.25	50.91
	80	99.67	0.17	99.36
	100	99.83	0.08	105.29
FV-USM	20	98.88	0.56	7.79
	50	99.49	0.25	19.88
	80	99.59	0.20	41.53
	100	99.79	0.10	56.10
SDUMLA-HMT	20	99.37	0.31	3.33
	50	99.84	0.08	7.82
	80	99.84	0.08	9.91
	100	99.84	0.08	12

**Table 8 sensors-25-02279-t008:** The recognition accuracy, EER value, and model training and classification time obtained by using the different momentum parameter in the adaptive threshold on the MMCBNU_6000 dataset. The best results are shown in bold.

Momentum Parameter	Accuracy (%)	EER (%)	Time (s)
0.5	99.67	0.17	503
0.6	99.67	0.17	516
0.7	99.67	0.17	504
0.8	99.67	0.17	470
**0.9**	**99.83**	**0.08**	459

**Table 9 sensors-25-02279-t009:** The recognition accuracy, EER value, and model training and classification time obtained by using the different threshold, as well as the addition of BN after each spiking convolutional layer on the FV-USM dataset. The best results are shown in bold.

Threshold	Accuracy (%)	EER (%)	Time (s)
Convt_1=1,Convt_2=1	99.59	0.24	232
Convt_1=2,Convt_2=1	99.49	0.24	231
Convt_1=3,Convt_2=1	99.49	0.24	222
Convt_1=4,Convt_2=1	99.39	0.31	224
AT	99.69	0.15	330
** AT+BN **	**99.79**	**0.10**	350

**Table 10 sensors-25-02279-t010:** The effects of lateral inhibition mechanisms on model recognition accuracy, EER, and model training and classification time evaluated on the MMCBNU_6000 dataset. The best results are shown in bold.

Model Configuration	Accuracy (%)	EER (%)	Time (s)
Baseline (no inhibition)	99.33	0.33	501
Conv1 potential inhibition	99.50	0.25	500
Conv2 potential inhibition	99.33	0.33	426
Conv2 spike inhibition	99.83	0.08	470
**Full inhibition scheme**	**99.83**	**0.08**	459

**Table 11 sensors-25-02279-t011:** The recognition accuracy (%), EER (%), parameter quantities, and FLOPs obtained by using different deep learning models on MMCBNU_6000 dataset.

	Accuracy (%)	EER (%)	Params ($\times 10^{3}$)	FLOPs ($\times 10^{6}$)
AlexNet	97.83	0.55	59,462	425
ResNet18	99.67	0.08	11,484	1150
ResNet34	99.33	0.17	21,592	2333
ResNet50	97.83	0.64	24,737	2619
VGG16	99.67	0.08	136,718	9461
DenseNet161	99.67	0.08	27,797	4634
CNN + CBAM	99.61	0.15	1261	13
**ATSNN**	**99.83**	**0.08**	**3**	**8**

**Table 12 sensors-25-02279-t012:** The recognition accuracy (%), EER (%), parameter quantities, and FLOPs obtained by using different deep learning models on FV-USM dataset.

	Accuracy (%)	EER (%)	Params ($\times 10^{3}$)	FLOPs ($\times 10^{6}$)
AlexNet	97.66	1.10	59,019	401
ResNet18	99.39	0.29	11,428	1226
ResNet34	99.28	0.34	21,537	2476
ResNet50	99.19	0.29	24,516	2778
VGG16	99.59	0.19	136,276	9086
DenseNet161	99.59	0.15	27,558	4482
CNN + CBAM	99.33	0.26	958	12
**ATSNN**	**99.79**	**0.10**	**3**	**2**

**Table 13 sensors-25-02279-t013:** The recognition accuracy (%), EER (%), parameter quantities, and FLOPs obtained by using different deep learning models on on SDUMLA-HMT dataset.

	Accuracy (%)	EER (%)	Params ($\times 10^{3}$)	FLOPs ($\times 10^{6}$)
AlexNet	99.05	0.24	59,462	521
ResNet18	99.21	0.20	11,484	1437
ResNet34	99.37	0.16	21,592	2916
ResNet50	98.74	0.32	24,737	3274
VGG16	99.05	0.24	136,718	11,796
DenseNet161	99.37	0.16	59,462	521
CNN + CBAM	99.37	0.16	1549	16
**ATSNN**	**99.84**	**0.08**	**3**	**2**

**Table 14 sensors-25-02279-t014:** EER (%) results of our ATSNN model compared with some state-of-the-art finger vein verification methods on the three finger vein datasets.

Category	Method	MMCBNU_6000	FV-USM	SDUMLA-HMT
Hand-crafted	SRLRR [54]	-	-	3.75
	LPQ [55]	1.00	1.92	-
	RLFs [56]	3.33	0.93	-
	PLS-DA [57]	0.63	0.15	2.15
Deep learning	Rec-FFVN [58]	-	2.17	3.30
	Semi-PFVN [58]	-	1.37	0.99
	Two-stream CNN [29]	0.13	-	1.26
	FCN [59]	-	1.42	-
	CNN-CO [60]	0.74	-	2.37
	CAE [61]	-	0.12	0.21
	Lightweight CNN [62]	0.50	1.07	-
	**ATSNN**	**0.08**	**0.10**	**0.08**

## Data Availability

The data presented in this study are openly available in [MMCBNU_6000], at https://doi.org/10.3390/s131114339. [FV_USM], at https://doi.org/10.1016/j.eswa.2013.11.033. [SDUMLA-HMT], at https://doi.org/10.1007/978-3-642-25449-9_33.

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
