# Peer review of "Finger Vein Recognition Based on Unsupervised Spiking Convolutional Neural Network with Adaptive Firing Threshold"

_sensors, 2025, doi:10.3390/s25072279_

Round 1

Reviewer 1 Report

Comments and Suggestions for Authors

This manuscript introduces a finger vein recognition model based on spiking convolutional neural network, incorporating discrete spike encodings of finger vein images to capture spatio-temporal dynamic features. While the work exhibits novelty, some crucial aspects need to be addressed for publication assessment:

1. Regarding the proposed ATSNN model, some crucial preprocessing steps are needed prior to spike encoding, including local normalization and pointwise inhibition, etc. Clarification is needed on these steps, why these steps are performed, what they do specifically, and a detailed description of these pre-processing steps. These details should be explained more comprehensively.

2. Also for the ATSNN model, although the dynamic threshold strategy is adopted, it is still necessary to set an initial threshold first, how the initial threshold is set, and whether different initial thresholds have certain effects on the model performance.

3. During the model training, it seems that only two spiking convolutional layers need to be trained, and the first layer has few training epochs (2 epochs), while the second layer has many epochs (20 epochs). How to determine the appropriate training epochs is a problem worth exploring.

4. I noticed that after updating the parameters of the spiking convolutional kernel at each layer, a lateral inhibition mechanism is incorporated to suppress the firing of neurons in other feature maps. It is suggested to analyze its function by experiment.

Reviewer 2 Report

Comments and Suggestions for Authors

This paper introduces an Adaptive Firing Threshold-based Spiking Neural Network (ATSNN) for FVR. The aim is to achieve more significant accuracy improvements in finger vein recognition (FVR) tasks. The main contribution of ATSNN is that it has an adaptive mechanism of dynamic adjustment, which has more advantages than traditional deep neural networks (DNNs). The advantages of ATSNN are its smaller number of parameters and model complexity, as well as its more biologically plausible.

In this paper, a strategy for calculating the adaptive threshold (AT) is mentioned in the Adaptive Threshold Strategy. The momentum parameter M is set to 0.9 in its calculation formula. It is suggested that the author can add theoretical explanations or experiments to prove its rationality.

When explaining STDP and Wins-Take-All (WTA) strategy, it is suggested that the authors add some mathematical derivation process of WTA and logical relationship between WAT and STDP. Perhaps a clearer theoretical analysis will make it easier for readers to understand.

In this paper, some of the references cited were published 5 years ago, and I suggest that the authors add some of SNN's latest research results in the field of biometric field, such as 

[1]Shen, J., Zhao, Y., Liu, J. K., & Wang, Y. (2021). HybridSNN: Combining bio-machine strengths by boosting adaptive spiking neural networks. IEEE Transactions on Neural Networks and Learning Systems, 34(9), 5841-5855. [2]Shen, J., Wang, K., Gao, W., Liu, J. K., Xu, Q., Pan, G., ... & Tang, H. (2025). Temporal spiking generative adversarial networks for heading direction decoding. Neural Networks184, 106975.[3] Xu, Q., Fang, X., Li, Y., Shen, J., Ma, D., Xu, Y., & Pan, G. (2024, October). RSNN: Recurrent Spiking Neural Networks for Dynamic Spatial-Temporal Information Processing. In Proceedings of the 32nd ACM International Conference on Multimedia (pp. 10602-10610). [4] Chen, K., Chen, S., Zhang, J., Zhang, B., Zheng, Y., Huang, T., & Yu, Z. (2025). Spikereveal: Unlocking temporal sequences from real blurry inputs with spike streams. Advances in Neural Information Processing Systems37, 62673-62696. [5] Bu, T., Li, M., & Yu, Z. (2024). Training-free Conversion of Pretrained ANNs to SNNs for Low-Power and High-Performance Applications. arXiv preprint arXiv:2409.03368.

In the experimental part, this paper conducts detailed experiments on the window size, the scale and orientation of Gabor kernels and DoG kernels, and the size of ROI image. It is suggested that while listing the experimental data, the authors should add more discussion about the biological plausibility of the model, such as how ATSNN has advantages in sparsity and event-driven compared with traditional DNN models.

Comments on the Quality of English Language

 The English could be improved to more clearly express the research.

Reviewer 3 Report

Comments and Suggestions for Authors

The author achieved excellent personal authentication with a very small number of parameters by processing finger vein images using SNN and SVM. Figures 9–11 are important, as they clearly demonstrate the superiority of SNN compared to deep learning, making this an outstanding research result. However, some parts lack sufficient information, making it difficult to understand the specific procedures. Before publication, the following additions are necessary.

  1. The specific usage of STDP is unclear. In Figure 3, "WTA" and "STDP" are mentioned after "AT". Section 2.2.2 describes "simplified STDP learning rules and a lateral inhibition mechanism." Which inter-neuron connections are updated by STDP? It is stated that the kernel parameters were updated using STDP. Does this mean that STDP was applied between the layers before and after the convolution, rather than the neurons inside the same layer?
  2. The significant reduction in the number of parameters while maintaining image recognition accuracy suggests that STDP plays a crucial role. It is necessary to discuss the role of STDP in this algorithm. STDP is typically used to enable the network to memorize frequently occurring spatiotemporal patterns, but what is its specific role in this case? How does it contribute to the reduction in the number of parameters?
  3. Based on the introduction, it appears that the classification of vein images was applied to personal authentication. However, I couldn’t find the explicit description neither in Sections 2.3 nor in the abstract. A clear explanation is needed regarding the specific classification task—was it solely personal authentication, or did it also classify finger types as well?
  4. The abstract states "capture the spatio-temporal dynamic features." But in reality, static images are converted into spatio-temporal patterns through spike coding, not capturing the spatio-temporal dynamic features of the original images. This is misleading and should be corrected.

Round 2

Reviewer 3 Report

Comments and Suggestions for Authors

My questions and comments are well addressed.